# Renoprotection of Selected Antioxidant-Rich Foods (Water Spinach and Red Grape) and Probiotics in Gentamicin-Induced Nephrotoxicity and Oxidative Stress in Rats

**DOI:** 10.3390/life12010060

**Published:** 2022-01-03

**Authors:** Sneha Sarwar, Md. Jamal Hossain, Nafis Md. Irfan, Tamima Ahsan, Md. Saidul Arefin, Arebia Rahman, Abdullah Alsubaie, Badr Alharthi, Mayeen Uddin Khandaker, David A. Bradley, Talha Bin Emran, Sheikh Nazrul Islam

**Affiliations:** 1Institute of Nutrition and Food Science, University of Dhaka, Dhaka 1000, Bangladesh; snehasarwar4@gmail.com (S.S.); nafis.irfan@du.ac.bd (N.M.I.); tamimaahsan@gmail.com (T.A.); arefin.saidul@gmail.com (M.S.A.); 2Department of Pharmacy, State University of Bangladesh, 77 Satmasjid Road, Dhanmondi, Dhaka 1205, Bangladesh; 3Interdisciplinary Graduate Program in Human Toxicology, University of Iowa, Iowa City, IA 52246, USA; 4Department of Internal Medicine, University of Iowa, Iowa City, IA 52246, USA; 5Department of Pathology, Dhaka Medical College and Hospital, Dhaka 1000, Bangladesh; dr.arebia@gmail.com; 6Department of Physics, College of Khurma, Taif University, P.O. Box 11099, Taif 21944, Saudi Arabia; a.alsubaie@tu.edu.sa; 7Department of Biology, University College of Al Khurmah, Taif University, PO. Box 11099, Taif 21944, Saudi Arabia; b.harthi@tu.edu.sa; 8Centre for Applied Physics and Radiation Technologies, School of Engineering and Technology, Sunway University, Bandar Sunway, Petaling Jaya 47500, Malaysia; mayeenk@sunway.edu.my (M.U.K.); d.a.bradley@surrey.ac.uk (D.A.B.); 9Department of Physics, University of Surrey, Guilford GU2 7XH, UK; 10Department of Pharmacy, BGC Trust University Bangladesh, Chittagong 4381, Bangladesh; talhabmb@bgctub.ac.bd

**Keywords:** nephroprotection, water spinach, red grape, probiotics, gentamicin-induced nephropathy, oxidative stress, nitrosative stress

## Abstract

Objectives: The current study investigated the curative effects of two selected antioxidant-rich foods (water spinach and red grape) and probiotics on the kidney exposed to nephrotoxicity induced by gentamicin. Methods: A total of 30 Wistar Albino female rats equally divided into six groups were studied for seven days. Except for the normal control (NC) group, all groups received 80 mg/kg/day gentamicin (GEN) injection intra-peritoneally for seven days. NC and GEN groups received only regular diet. In the water spinach group (GEN + WS) and red grape (GEN + RG) groups, rats were provided with 20 g/rat/day of boiled water spinach and 5 mL/rat/day of red grape juice, respectively. The probiotic (GEN + P_4_) and (GEN + P_8_) groups received 4 × 10^9^ and 8 × 10^9^ viable bacteria, respectively. On the 8th day, all the rats were sacrificed to collect blood and kidney. Serum creatinine, urea, uric acid, malondialdehyde (MDA), nitric oxide (NO), and superoxide dismutase (SOD) were analyzed. In addition, kidney histopathology was taken for final observation. Results: Both antioxidant-rich foods and probiotic (P_4_) significantly (*p* < 0.05) attenuated the GEN-induced oxidative and nitrosative stress and improved kidney function by lowering uremic toxin (serum creatinine, and uric acid) levels. Histopathological findings of kidney tissues of all groups were consistent with the biochemical findings. Conclusion: The current preclinical study suggests that the consumption of antioxidant-rich foods might be a promising fighting option against gentamycin-induced nephrotoxicity and oxidative stress. However, extensive studies and clinical monitoring are immediately required to determine the appropriate probiotic doses and mechanism of action for such effects.

## 1. Introduction

Nephrotoxicity can be defined as any type of renal injury mediated directly or indirectly by drugs (overdose, drug–drug interactions, or adverse effects), acute kidney failure, tubulopathies, obesity, diabetes, hypertension, and so on. Alarmingly, nearly 60% of the hospitalized patients in the intensive care unit are suffering from acute kidney injury (AKI), where the drug-induced nephrotoxicity is responsible for around 15% of the cases, holding a third leading cause for AKI [1]. In addition, chronic inflammation and oxidative stress play a vital role in kidney diseases and their complications [2]. Therefore, it is an urgent necessity to generate a mitigating strategy of nephrotoxicity.

Gentamicin, an antibiotic of aminoglycoside group, is preferably used against Gram-negative bacteria. Nephrotoxicity is one of the major adverse effects of gentamicin administration, and gentamycin-induced nephrotoxicity is well-known [3,4]. Gentamicin usually accumulates within the renal tubules and lysosomes [5]. Several pieces of evidence revealed that lipid peroxidation and nitric oxide generation are elevated within the renal cortex following the administration of gentamicin [6,7]. Besides, the production of excessive reactive oxygen species (ROS) causes this nephropathy [8]. Both glomerular and tubular injuries caused by this gentamicin can simulate the complications of nephropathy [9]. Nephrotoxicity leads to the accumulation of uremic toxins within the body. Uremic illness is caused by the excessive deposition of organic waste products that are typically removed by the kidneys. Uremia is linked with abnormalities in the gastrointestinal mucosa and disequilibrium in the intestinal ecosystem [10]. Toxins in the body increase the pH of the blood, cause an increase in pathobionts, and a decrease in the healthy gut microbiome. The pathobionts, through inflammatory reactions, disrupt cellular integrity and enter the gut. They produce more uremic toxins in the gut and thus lower the number of healthy gut microbiomes [11].

Currently, phytochemicals or herbal drugs derived from various natural sources, including several functional foods, are commonly used against so many disorders, and this healing strategy is growing popularity day by day worldwide [12]. Many natural products are believed to have potent nephroprotective effects, and many formulations of phytochemicals, plant extracts, animal extracts, herbals, and so on have been reported to have renoprotection against drug-induced kidney diseases [13]. Several experimental studies have shown the antioxidant potential of red grape and water spinach due to the presence of polyphenolic compounds in these products. The polyphenolic compounds are vital for the free-radical scavenging and regeneration of renal tubules. These beneficial effects may provide nephroprotective potentialities [14,15]. Although the kidney protective effects of water spinach are less understood from the current evidence, the red grapes showed promising results against several drug-induced renal toxicities [13]. Safa et al. [16] reported that red grape seed extract showed nephroprotection against gentamicin-induced acute kidney injury. Besides, many studies have reported promising renal protective effects of resveratrol found in red grape [13,17,18,19].

According to the definition of the World Health Organization (WHO) and Food and Agricultural Organization (FAO), probiotics are well-defined as living microorganisms, which confer a benefit on the host when given in adequate quantities [20]. Probiotics can be used as dietary supplements and drugs. The most common probiotics are *Lactobacilli* and *Bifidobacterium*. Different species are present in the normal microbiota of animal and human gastrointestinal and genitourinary tracts and confer many health benefits [21]. Besides, probiotic formulations might help in the restoration of healthy gut microbes and have the potentiality to degrade the accumulated uremic waste [22]. Hence, they can be promising solutions in resolving nephrotoxicity by lowering uremia.

In this present research, both approaches (either by scavenging free radicals or by degrading accumulated waste) to curb renal disease have been studied. Several previous studies [14,15,16,17,18] reported the drug-induced renoprotection of the extracts of different parts of red grape and water spinach or isolated compounds from these sources. However, to the best of our searching experiences, this is the first study where we used red grape and water spinach in their consumable forms (red grape juice and boiled water spinach) and probiotics due to potent antioxidant properties to treat nephrotoxicity. Therefore, this study aimed to evaluate the nephroprotective role of the selected antioxidant-rich foods (red grape and water spinach) and probiotics in gentamicin-induced nephropathy in animal models.

## 2. Materials and Methods

### 2.1. Collection and Preparation of Food Samples

Food samples were selected based on their values of certain nutrients (a rich source of total polyphenol content, antioxidant, vitamin C, and beta carotene). This primary selection was made by searching database of the Food composition table for Bangladesh [23] and the Food composition table for India [24]. From these databases, one fruit sample: Red grape (*Vitis vinifera*) and one vegetable sample: water spinach (*Ipoema aquatic)* were chosen for the investigation. Both of the samples were purchased from Sutrapur Bazar, Old Dhaka, Bangladesh. The leaves of water spinach were finely washed, chopped, and boiled. In a stainless-steel pan, 100 g water spinach was boiled in 150 mL of water and cooked until it became tender [25]. No water needed to be discarded as all water dried during gently heating. The boiled sample was mixed with the regular rat diet at 4:1 ratio and then administered to the rats. Red grapes were washed properly, and then the whole fruit was strained accurately, and the juice was given by tube feeding.

### 2.2. Collection and Preparation of Probiotics

A commercial probiotic formula, PROBIO, manufactured by Square Pharmaceuticals Ltd. Bangladesh, was used. Probiotics were purchased from an authentic drug store in Dhaka (the Capital city of Bangladesh). It contains in total 4 × 10^9^ CFU/unit live cultures, consisting, *Lactobacillus acidophilus* (2 × 10^9^), Lactobacillus bulgaricus (1 × 10^9^), Bifidobacterium bifidum (1 × 10^9^) fructo-oligosaccharide (100 mg) as prebiotic. Then, the capsules were mixed with water and were fed to the rats. 

### 2.3. Animal Collection and Acclimatization

The experiment was carried out in the Institute of Nutrition and Food Science (INFS) in collaboration with the Department of Pathology of Dhaka Medical College and Hospital and Sir Salimullah Medical College and Mitford Hospital (SSMC) Dhaka. Due to the unavailability of male rats during the study period in the lab, thirty (30) female Wister Albino rats were collected from the animal house of the Pharmacy Department, Jahangirnagar University, Savar. Their age was between 60–70 days, and their weight 200–250 g The rats were kept in the animal house of the Institute of Nutrition and Food Science (INFS) of the University of Dhaka. The rats were housed in metallic cages in a temperature-controlled room (21 ± 2 °C) with a 12-h light/dark cycle. The room was well ventilated and properly sanitized. The rats were acclimatized for 14 days. During this time, they were fed ad libitum with the standard laboratory diet (Table 1).

All the experiments in this study were carried out according to the animal research guidelines and pre-conditions attributed by the World Medical Association statement on the use of animals in biomedical research. Besides, the Animal Ethics Committee, Institutional Ethical Review Committee, State University of Bangladesh (SUB-IERC) has critically reviewed and approved the detailed protocols of the study and provided ethical permission for conducting the research with an approval number (2021-05-10/SUB/A-ERC/0008).

### 2.4. Drugs and Chemicals

Commercial Gentamicin injection (80 mg) was collected from a model pharmacy (Dhaka, Bangladesh). Laboratory standard Thiobarbituric acid, Trichloroacetic acid, Hydrochloric acid, 10% formalin, 0.9% saline, Phosphate buffer, Detergent, Uricase, Dichlorophenol sulphonate, Ascorbate oxidase, Peroxidase, Amino-antipyrine, Urease Enzyme reagent, EDTA, Sodium salicylate, Sodium nitroprusside, Alkaline hypochlorite, Sodium hypochlorite, Sodium hydroxide, Urea standard, Picric acid, Creatinine Standard, Nitrite standard solution, Griess reagent, and Alcohol were purchased from Sigma Aldrich (St. Louis, MO, USA) and used during the research.

### 2.5. Experimental Design

After the acclimatization period, rats were divided randomly into six equal groups, including five animals in each group. The study period was seven days. All the animals except the normal control group were exposed to nephrotoxicity by administering gentamicin (GEN) 80 mg/kg/day intraperitoneally throughout the study period [26]. Groups were as follows:Normal control (NC) group: This group of rats received no treatment/gentamicin and was given the standard laboratory diet.Gentamicin (GEN) group: Rats were provided with the standard laboratory diet.Water Spinach (GEN + WS) group: In this group, the laboratory diet was mixed with boiled water spinach in a 1:4 ratio (laboratory diet:WS). Each rat received about 20 g boiled water spinach per day for seven days.Red Grape (GEN + RG) group: Apart from the basal diet, each rat received 5 mL of red grape juice for the entire study period (7 days). The juice was provided through oral gavage feeding tubes to ensure each rat’s similar amount of intake.Probiotic 4B (GEN + P_4_) group: Throughout the entire study week, 4 × 10^9^ CFUs were provided through oral gavage feeding tubes to each rat each day.Probiotic 8B (GEN + P_8_) group: For the whole study period, 8 × 10^9^ CFUs were given through feeding tubes to each rat per day.

### 2.6. Blood Sampling

On the 8th day, all rats were anesthetized with the help of chloroform (30%) and sacrificed. From all rats, blood samples of approximately 3 mL were collected from the heart using sterile disposable syringes and were taken in tubes with valid identification numbers. After 30 min, blood samples were centrifuged at a rate of 3000 rpm for 15 min. Following that, supernatant serum was collected in a labeled Eppendorf tube and stored in the refrigerator at −18 °C for determination of plasma creatinine, urea, uric acid, malondialdehyde (MDA), Nitric Oxide (NO) levels, and Superoxide Dismutase (SOD) activities.

### 2.7. Tissue Sampling from Kidney

After collecting blood samples, all the rats were killed, and both kidneys were collected from each of the rats by meticulous dissection method, washed in 0.9% NaCl saline, and wiped in tissue paper. Then, both kidneys were weighed, and the mean weight was recorded. The weight of the kidney was measured by an electric balance analyzer (Mettler Toledo, Zurich, Switzerland) and preserved in 10% neutral buffered formalin and stored at −18 °C. Then, the right kidney from each rat was selected for histological investigations.

### 2.8. Estimation of Renal Function

The kidney function was estimated by measuring serum creatinine, urea, and uric acid. These parameters of rats were analyzed with a semi-auto biochemistry analyzer (AUSTRIA). Serum creatinine was determined by the Kinetic method without deproteinization–Jaffle Reaction [27] and expressed as mg/dL. Uric acid was determined by liquid uric acid uricase-PAP for in vitro diagnostic use only [28]. Serum urea was quantitatively estimated by Urease/Salicylate Colorimetric method endpoint [29]. Both serum uric acid and serum urea were expressed as mg/dL during statistical analysis.

### 2.9. Estimation of Stress Parameters

Serum Malondialdehyde (MDA) level, Superoxide Dismutase (SOD), and serum nitric oxide level (NO) were analyzed by a semi-auto analyzer. Estimation of MDA was done by the thiobarbituric acid assay method [30]. Serum Cu-Zn containing superoxide dismutase activity was enumerated spectrophotometrically [31]. Serum NO level was assessed by the method described by Menaka et al. [32].

### 2.10. Preparation of Kidney Specimen for Histopathological Examination

The collected and appropriately stored kidneys were processed and embedded in paraffin wax and sections were taken using a microtome. Sections were then stained with haematoxylin and eosin and examined under light microscope [33].

### 2.11. Photography

Photographs were taken from the representative sections of each group using a camera fitted with the microscope.

### 2.12. Statistical Analysis

The obtained data were statistically analyzed, and the results were expressed as means ± standard deviation (SD). Differences between groups were assessed using a one-way analysis of variance (ANOVA). The Tukey test was conducted for multiple comparisons when group deference was significant. Paired *t*-test was conducted to observe the significant bodyweight variation due to the administration of drugs. Results were considered statistically significant at *p* < 0.05. All the analyses were run using the SPSS 20.0 for Windows.

## 3. Result

### 3.1. Effects of Gentamicin on Bodyweight and Kidney Weight

Bodyweight variation was observed by comparing the bodyweights measured at the initiation and termination of the study for groups of rats, including the normal control group. As illustrated in Table 2, the average body weight in each group before the initiation of the study was found to be around 196 g. It is clear from Table 2 that the GEN group of rats significantly lost their bodyweight (mean difference = −14.42; *p* = 0.004). Besides, the high-dose of probiotics acted synergistically to lose body weight of rats (mean difference of GEN + P_8_ = −21.2; *p* = 0.004). On the other hand, water spinach, red grape, and lower dose probiotic protected bodyweight loss due to gentamicin-induced effects (positive mean difference for all three groups). Besides, the normal control of rats showed significantly elevated bodyweight (mean difference = 13.4; *p* < 0.001).

The weight of the kidneys was determined to identify any significant change in kidney weight due to toxicity and intervention. Chemical infusion in the body caused a rise in kidney weight. As shown in Figure 1, kidney weight was significantly lower in the normal control group and the GEN + P_4_ group compared to the GEN group. However, alterations in kidney weight in the other treatment groups were not significant.

### 3.2. Effects of Gentamicin on Biochemical Parameters

The renal function of the rats was assessed at the end of the study through measuring three biochemical parameters (serum creatinine, uric acid and urea). Serum creatinine level (mean ± SD) was significantly lower for water spinach (GEN + WS: 0.82 ± 0.28 mg/dL) and normal control (0.67 ± 0.15 mg/dL) groups compared to the GEN group (1.64 ± 0.55 dL) of rats (Table 3). In case of serum urea level, normal control group (17.94 ± 5.40 mg/dL) and 4 billion of probiotics (GEN + P_4_: 17.94 ± 3.01 mg/dL) showed significantly (*p* < 0.05) lower concentration than the only GEN group (82.57 ± 44.28 mg/dL) of rats. However, the higher dose of probiotics (8 × 10^9^) group (GEN + P_8_: 116.74 ± 11.76 mg/dL) showed significantly higher serum urea level compared to the only GEN control group (82.57 ± 44.28 mg/dL). Notably, all the groups (normal control: 7.31 ± 4.12 mg/dL, GEN + WS: 10.22 ± 1.91, GEN + RG: 7.47 ± 3.46 mg/dL, GEN + P_4_: 5.42 ± 1.52 mg/dL, GEN + P_8_: 7.43 ± 1.27 mg/dL) showed significantly (*p* < 0.05) lowered uric acid level than the GEN control (17.77 ± 1.91 mg/dL) group (Table 3).

Moreover, a significant alteration is recognized in several stress biomarkers in serum. The MDA level was significantly (*p* < 0.05) lower in all the treatment groups (2.5 ± 0.402 to 5.46 ± 2.84 nmol/L), including the NC group (3.86 ± 1.34 nmol/L) than the GEN group (12.76 ± 1.52 nmol/L) (Table 3). Similarly, all the groups exerted significantly (*p* < 0.05) lowered serum NO level (1.08 ± 0.21 to 1.82 ± 0.33 nmol/mL) compared to the GEN group of rats. Furthermore, the SOD level was also higher in all groups compared to the GEN group. However, no significant difference was observed in the case of serum SOD level among the treatment groups and gentamicin control group.

### 3.3. Histopathology Analysis

The observations from the histopathology analysis of the kidney tissues were consistent with the results of the biochemical parameters. In Figure 2A, the NC group’s kidney tissues showed normal glomerular structure and interstitium and no inflammatory cell infiltration. The kidney tissues of the GEN group exhibited extensive necrosis, glomerular congestion, interstitial congestion, and inflammatory cell infiltration (Figure 2B). The kidney tissues, glomerular and interstitial infiltration of inflammatory cells of GEN + WS group were focally present along with mildly congested blood vessels in the interstitium, mild hyelin cast in the tubule (Figure 2C). The kidney tissues of the Red Grape (GEN + RG) group revealed mild desquamation of proximal tubules. Focal glomerular and interstitial infiltrations of inflammatory cells along with mildly congested blood vessels in the interstitial mild hyelin cast in the tubule were found (Figure 2D). The probiotic 4B (GEN + P_4_) Group showed no glomerular and interstitial infiltration of inflammatory cells along with mildly congested blood vessels in the interstitium (focally present) (Figure 2E). The probiotic 8B (GEN + P_8_) group (Figure 2F) exhibited necrosis of proximal tubules. It also revealed no glomerular and interstitial infiltration of inflammatory cells along with some congested blood vessels in the interstitium and severe hyelin cast in the tissue. It also showed an increased number of glomerulus with increased size.

## 4. Discussion

The present study revealed the effects of red grapes and boiled water spinach as potential dietary alternatives to curb renal failure. The study has also provided empirical pieces of evidence for two different doses (4 × 10^9^ and 8 × 10^9^) of the same probiotic formulation (containing 2 *Lactobacillus* strains and a *Bifidobacterium* strain) on nephrotoxicity and oxidative stress in the animal model. Animal studies prior to human trials provide evidence on efficacy and minimize the unwanted risk of any intervention. For nephrotoxicity studies, rats are mainly used [34]. Therefore, this study was conducted on Wistar Albino rats, where nephrotoxicity was induced by intraperitoneal injection of gentamicin [33,35].

Generally, the underlying causes of kidney problems are low glomerular filtration rate, irregularities in kidney metabolism, tubular function, or oxidative stress [36]. Molecular-level alterations of the kidney tissue occur even before nitrogenous compounds, for example, serum urea, creatinine, accumulate in systematic circulation. Kidney proximal tubules contain ample mitochondria that are critical for the energy production process of reabsorption of water molecules and solutes. It is very well-known that mitochondria are the largest producers of reactive oxygen species (ROS), which augment the susceptibility of kidney injuries through oxidative stress. ROS and prooxidants generated during acute or chronic kidney disorders may further reinforce the severity of the disease and contribute to the pathogenesis of the subsequent complications [37]. Oxidative stress is increased in patients with kidney disorders due to increased oxidant activity and reduced antioxidant capacity, leading to renal dysfunction [38]. Thus, it was anticipated that reversing the oxidative stress condition might improve kidney function. Moreover, renal dysfunction leads to the accumulation of waste materials within the body, increasing oxidative stress which further aggravates kidney dysfunctionality [37,38]. Therefore, disposal of the waste might be another way of reversing the dysfunctional kidney situation.

In this study, GEN administration produced significant nephrotoxicity, confirmed by a marked increase in stress and renal function parameters and histological changes. GEN antibiotics remain in contact with renal proximal tubular cells due to renal reabsorption and prolonged concentration processes. Then, the drug potentially causes damage to the tubular transport system through induction of oxidative stress and finally leads to severe damage of mitochondria of the tubule [39]. Besides, GEN mediates the generation of reactive oxygen or nitrogen species (ROS or RNS) and decreases the activity of endogenous antioxidants (e.g., SOD) [40]. Accumulation of ROS leads to a loss of intracellular organelles, followed by other structural and functional alterations of the renal tissues, which causes GFR disruption and acute tubular necrosis [41].

Moreover, the physiological changes and several alterations in the anatomy have been observed in the GEN group during the bodyweight variation test. The percent of weight reduction in the GEN group was significantly higher than any other group. It might be happened due to the consequence of skeletal muscle breakdown and declined lean body mass [42]. Apart from this, we have recognized a significant increase in kidney weight, which has been noticed in the GEN group compared to the normal control’s kidney. Kidney weight was decreased in all the treatment groups, but the decrease was significant in the GEN + P_4_ group. Accumulation of toxic elements due to the GEN toxicity might be a reason for this increased weight; on the contrary, the removal of the toxic materials from the kidney might induce weight normalization [43].

The study’s primary objective was to investigate the reno-protective property of the selected food items and probiotics by delaying or lowering the progression of oxidative stress. Boiled water spinach and red grape consumption have positively influenced renal and stress function parameters in this research. The current findings demonstrated the reno-protective and antioxidative properties of both functional foods against gentamicin-induced renal toxicity. Both foods may be attributed to this nephroprotective potentiality due to the presence of phenolic compounds and antioxidative properties [13,44,45]. Biochemical analysis of water spinach and red grapes extracts showed the presence of phenols, flavonoids, tannins, ß- carotene, and soluble carbohydrate, which have diversely protective effects against various toxicity, including renal injuries [46,47]. Besides, the potent antioxidant compound resveratrol is present in red grape, which is highly effective against various drug-induced nephrotoxicity [13,48]. Resveratrol in red grape might have lowered oxidative stress and renal dysfunction by increasing nitric oxide synthesis and superoxide dismutase expression [49]. Because of the interruptions of the pathway how the inflammatory-oxidative stress is associated with kidney complications, resveratrol might contribute to controlling the chronic kidney disease-related metabolic derangements. Thus, the agent mitigates oxidative stress-induced kidney damage and restores renal functionality [50]. This is also supported by the current findings, as improved oxidative stress and kidney function markers were observed simultaneously.

Moreover, boiling water spinach increases the bioavailability of the phenolic compounds that might boost the antioxidant potentiality of the water spinach [25]. Water spinach contains ascorbic acid, gallic acid, chlorogenic acid, myricetin, quercetin, and apigenin which are thought to prevent apoptosis, thus inhibiting oxidative stress [14,45,51]. The current findings of this research are in harmony with the previous reports where the authors reported the improved renal function upon water spinach and red grape or resveratrol ingestion [52,53]. Therefore, according to the findings of this study, the red grape and water spinach in their regular consumable form might be an excellent way to reverse oxidative stress, hence providing renoprotection.

Another important aim of the study was to investigate the degradation of the accumulated waste within the body with the help of probiotic formulation that might lower oxidative stress and provide nephroprotection. In this study, there was a significant reduction in mean serum uric following the treatment with probiotics (GEN + P_4_ and GEN + P_8_) than the GEN group. However, probiotics could not significantly decrease the serum creatinine level. As bacterial cells consume nitrogenous wastes (such as urea, uric acid, and creatinine) as an energy source for their growth and metabolism, therefore, probiotics diminish the concentration of these toxins in the circulation of uremic patients and contribute to the amelioration of biochemical imbalance [54,55]. The current findings are also consistent with other previous studies where probiotics were used for the improvement of renal function parameters [56,57]. It might be underlined that the effects of probiotics were in binary directions in two probiotic groups in case of urea level alterations. Probiotics would reduce urea levels because some of them synthesize urease enzymes that convert urea into the gut lumen, thereby reducing the plasma levels of urea [55]. The group which received the lower dose of probiotic (GEN + P_4_) validated this hypothesis both biochemically and histopathologically in this study. However, the GEN + P_8_ group had elevated the plasma level of urea, and the histopathology exhibits the presence of crystals in the renal tissues in this investigation. In kidney disease, a high amount of urea is diffused in the gastrointestinal tract (GIT). Some probiotics and pathobionts secrete the urease enzyme, which hydrolyzes the urea and thereby resulting in the generation of larger quantities of ammonia (NH_3_) and ammonium hydroxide (NH_4_OH). Thus, the pH is increased in the gut area and disrupts tight epithelial junction. Thus, the entry of pathobionts is promoted within the gut. In the gut, they convert all available sources of proteins to uremic retention solutes and aggravate inflammation [58,59,60]. This might be the underlying reason for increased urea in the GEN + P_8_ group. In several studies, this inconsistency has been reported; some have reported that probiotic supplementation decreased urea [11,55].

Moreover, probiotics lowered the oxidative stress groups by significantly reducing the level of MDA and NO. Even under the microscope, there was mild to no infiltration of inflammatory cells in the renal tissues. Probiotics may modulate the host’s ROS system either by their metal chelating ability with a metal chelator or by producing their antioxidant systems (such as SOD), or by altering intestinal microbiota [61]. Similar studies revealed that probiotics could inhibit ROS production and lower oxidative stress by their antioxidative properties [62,63].

### 4.1. Strengths and Limitations

The study’s major strength is using two different approaches (food trial and probiotic trial) to curb renal disease. The foods were used in their regular consumption form, not as an extract. The dose responsiveness of the probiotic was studied by using two different doses. Both biochemical and histopathological studies were conducted to see the effectiveness of the interventions.

There were several limitations of the study, such as the study period being short (i.e., only seven days excluding the acclimatization period). Having a significant anatomical change within this period is not possible. Besides, the urine sample was not analyzed. Although an elevation in the urea level has been spotted, the reason behind the elevation cannot be identified surely as a mixed probiotic formulation had been used. It can be due to the excessive amount of one specific strain or the high amount of the complete dose.

### 4.2. Future Research

Red grape and water spinach can be used as nephroprotective agents in nephrotoxicity. However, the dose responsiveness of these foods used should be studied rigorously. In terms of probiotic use, further extensive researches should be conducted as probiotics can function through various pathological means and affect the gut environment. The effective dose of probiotics should be investigated as two doses provided results in contradictory manners. In place of mixed strains of bacteria, a single specific strain can be studied. On an effective dose, probiotics can open a new window in curbing nephrotoxicity. Effective and economic nephroprotective agents can be obtained through further experiments. Further studies, including pharmacokinetics and pharmacodynamics investigations, are warranted to develop useful nephroprotective agents for humans.

## 5. Conclusions

The present study has been carried out to observe the nephroprotective effect of antioxidant-rich selected foods (red grape and water spinach) and probiotics (4 × 10^9^ and 8 × 10^9^ doses) against gentamicin induced-nephropathy and oxidative stress in the animal model. The red grape, water spinach, and low dose probiotic supplementation have exhibited remarkable nephroprotection by reducing oxidative stress and diminishing uremic toxins. These findings were evident from the histopathological and biochemical analysis. However, higher dose probiotic increased serum urea level. The study findings indicated that antioxidant-rich foods and probiotics on effective dosage can be promising solutions in curbing renal injury. Further studies should be carried out to reveal the underlying mechanisms of such activities.

## Figures and Tables

**Figure 1 life-12-00060-f001:**
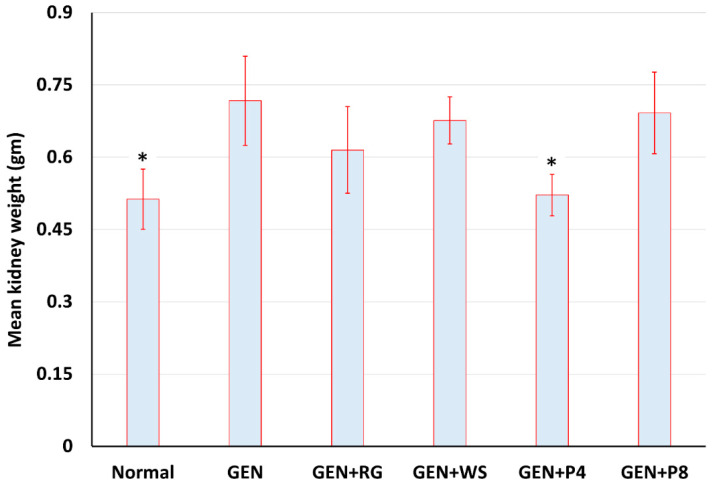
Change in mean kidney weight. NC = Normal Control, GEN = Gentamicin control group. GEN + RS = Red Grape, GEN + WS = Water spinach. GEN + P_4_ = Probiotic 4 × 10^9^, GEN + P_8_ = Probiotic 8 × 10^9^. The values were compared against GEN group, *p* < 0.05 is signified by (*).

**Figure 2 life-12-00060-f002:**
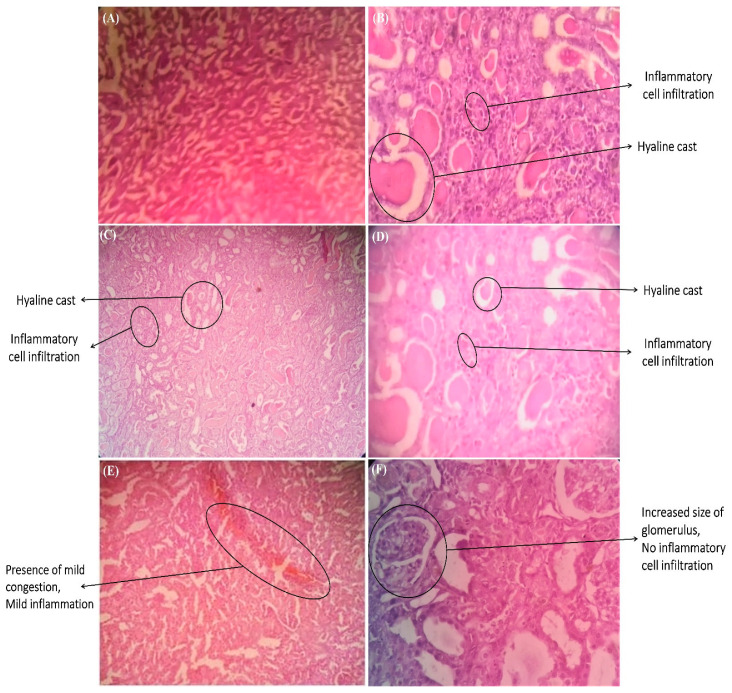
Kidney cell histopathology: (**A**) Normal control (NC) group, (**B**) Gentamicin control (GEN) group, (**C**) Water spinach (GEN + WS) group, (**D**) Red Grape (GEN + RG) group, (**E**) Probiotic 4B (GEN + P_4_) group, and (**F**) Probiotic 8B (GEN + P_8_) group.

**Table 1 life-12-00060-t001:** Composition of standard laboratory diet used in the study.

Ingredients	Percentage (%)
Rice polish and ground maize (*w*/*w*)	55.7
Ground whole grain wheat (*w*/*w*)	30.9
Dried fish meal (*w*/*w*)	5.4
Soybean oil (*v*/*w*)	3.2
Salt (NaCl) (*w*/*w*)	2.4
Vitamin mixture (Vitamin B_12_ and Vitamin C) (*w*/*w*)	0.4
Distilled water for mixing	2.0

**Table 2 life-12-00060-t002:** Effects of gentamicin, water spinach, red grape, and probiotics on body weight of rats. The bold format indicates the significant level (*p* < 0.05).

Groups	Initial Weight (g) Mean ± SD	Final Weight (g) Mean ± SD	Mean Difference (Final-Initial)	Paired *t*-Test	*p*-Value
Normal control (NC)	196.4 ± 52.22	209.80 ± 54.21	13.4	−11.49	<**0.001**
Gentamicin (GEN)	196.8 ± 37.11	181.98 ± 35.69	−14.42	6.158	**0.004**
Water spinach (GEN + WS)	196.4 ± 16.82	203.21 ± 12.05	6.81	−2.369	0.077
Red grape (GEN + RG)	196.0 ± 22.67	201.24 ± 21.57	5.24	−1.414	0.230
Probiotic (4B) (GEN + P_4_)	196.0 ± 12.20	207.0 ± 13.30	11.0	19.038	<**0.001**
Probiotic (8B) (GEN + P_8_)	196.0 ± 12.25	174.80 ± 13.72	−21.2	−5.982	**0.004**

**Table 3 life-12-00060-t003:** Alterations in kidney biochemical parameters among various groups of rats treated with gentamicin, water spinach, red grape, and probiotics.

Biochemical Parameters	Groups					
	Normal Control (NC)	Gentamicin Control (GEN)	Water Spinach (GEN + WS)	Red Grape (GEN + RG)	Probiotic (4B) (GEN + P_4_)	Probiotic (8B) (GEN + P_8_)
Creatinine (mg/dL)	0.67 ± 0.15 *	1.64 ± 0.55	0.82 ± 0.28 *	1.16 ± 0.44	1.48 ± 0.144	1.51 ± 0.31
Urea (mg/dL)	17.94 ± 5.40 *	82.57 ± 44.28	52.35 ± 16.22	81.08 ± 5.04	17.94 ±3.01 *	116.74 ± 11.76 *
Uric acid (mg/dL)	7.31 ± 4.12 *	17.77 ± 5.70	10.22 ± 1.919 *	7.47 ± 3.46 *	5.42± 1.52 *	7.43 ± 1.27 *
MDA (nmol/L)	3.86 ± 1.34 *	12.76 ± 1.52	4.03 ± 0.15 *	5.46 ± 2.84 *	4.54 ± 1.61 *	2.50 ± 0.402 *
NO (mmol/mL)	1.82 ± 0.33 *	3.42 ± 1.03	1.24 ± 0.21 *	1.50 ± 0.43 *	1.08 ± 0.21 *	1.36 ± 0.07 *
SOD (U/mL)	3.22 ± 1.73	1.68 ± 0.84	2.73 ± 0.79	2.98 ± 1.08	3.23 ± 0.485	3.25 ± 0.16

Values are expressed as Mean ± SD; * Significance level *p* < 0.05.

## Data Availability

All the obtained data of this research are presented in the manuscript.

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
