# Peer review of "Renoprotection of Selected Antioxidant-Rich Foods (Water Spinach and Red Grape) and Probiotics in Gentamicin-Induced Nephrotoxicity and Oxidative Stress in Rats"

_life, 2022, doi:10.3390/life12010060_

Round 1

Reviewer 1 Report

Comments to the authors:

Overall, this is a clear, concise, and well-written manuscript. Sufficient information about the previous study findings is presented for readers to follow the present study rationale and procedures. The article can be published with minor correction. The author is required the following corrections:

  1. The key objectives should be focused nicely in the 'Introduction' section. Literature search can be improved on the basis of previously published results. Try to include the existing research limitations also, how the present research unravels those limits.
  2. In the whole manuscript, summary of selective few points are written by considering few relevant papers. Few points were just written without well explanation. Discussion of data should be related to other similar species or previous study.
  3. Add a focus point in abstract. The abstract is long.
  4. Mark the inflammation in figure 2.
  5. Paired t-test was conducted to observe the significant bodyweight variation due to the administration of drugs. Is it necessary?
  6. Conclusion should concise based on the major findings.
  7. Few grammatical errors are observed throughout the article which need to be corrected.
  8. Arrange the references according to the journal guidelines (some used abbreviation)

Author Response

Reviewer 1

Comments and Suggestions for Authors

Comments to the authors:

Overall, this is a clear, concise, and well-written manuscript. Sufficient information about the previous study findings is presented for readers to follow the present study rationale and procedures. The article can be published with minor correction. The author is required the following corrections:

Authors’ responses:

Thank you very much for reviewing our manuscript and providing your expert opinions on our manuscript. We are grateful to you for recommending our manuscript for publication. We have carefully considered all of your comments and tried our level best to improve our manuscript according to your suggestions.

  1. The key objectives should be focused nicely in the 'Introduction' section. Literature search can be improved on the basis of previously published results. Try to include the existing research limitations also, how the present research unravels those limits.

Authors’ responses:

Thank you very much for the point regarding the introduction. We have carefully revised our introduction part and improved according to your suggestions. We have searched all the databases to improve the literature parts of introduction part of the manuscript (Lines: 84-89) and (Lines: 96-98). Also, the lines 99-108 revealed the novelty and clear objectives of the research.

2. In the whole manuscript, summary of selective few points are written by considering few relevant papers. Few points were just written without well explanation. Discussion of data should be related to other similar species or previous study.

Authors’ responses:

Thank you very much for the point. We have carefully revised the discussion point and correlated with the previous studies. We have justified all the findings of our study with the previous literature.

3. Add a focus point in abstract. The abstract is long.

Authors’ responses:

Thank you very much for your comments. We have revised and shortened the abstract with a focus point (Lines: 38-42).

4. Mark the inflammation in figure 2.

Authors’ responses:

Thank you very much. We appreciate your nice suggestion. We have marked the inflammation in figure 2.

5. Paired t-test was conducted to observe the significant bodyweight variation due to the administration of drugs. Is it necessary?

Authors’ responses:

Thank you very much for your question. Yes, it is necessary to see if bodyweight variation was significant or not due to administration of gentamicin. Bodyweight variation was observed by comparing the bodyweights measured at the initiation and termination of the study for groups of rats, including the normal control group (Lines: 213-215).

6. Conclusion should concise based on the major findings.

Authors’ responses:

Thank you very much for your suggestion. We have revised, improved and shortened the conclusion in a concise manner as follows:

The present study has been carried out to observe the nephroprotective effect of antioxidant-rich selected foods (red grape and water spinach) and probiotics (4 billion and 8 billion doses) against gentamicin induced-nephropathy and oxidative stress in the animal model. The red grape, water spinach, and low dose probiotic supplementation have exhibited remarkable nephroprotection by reducing oxidative stress and diminishing uremic toxins. These findings were evident from the histopathological and biochemical analysis. However, higher dose probiotic increased serum urea level. The study findings indicated antioxidant-rich foods and probiotics on effective dosage can be promising solutions in curbing renal injury. Further studies should be carried out to reveal the underlying mechanisms of such activities. (Lines: 411-422)

7. Few grammatical errors are observed throughout the article which need to be corrected.

Authors’ responses:

Thank you very much for suggestion. We have carefully revised the whole manuscript and eliminated all grammar and unintentional typos errors.

8. Arrange the references according to the journal guidelines (some used abbreviation)

Authors’ responses: Thank you very much for your comments. We have revised the referencing section and corrected according to the journal guidelines.

Reviewer 2 Report

The manuscript focuses on the nutritional benefits of red grapes, water spinach and certain probiotics against gentamicin induced nephrotoxicity. The objectives of the study are significant with regard to the emerging interest in  nutraceuticals or functional foods in therapeutic strategies or adjuvant therapies.

1.The minor issue in the manuscript is some grammatical errors; for example- 

In Material and methods, line 117....Remove "the" before "searching"

In animal collection & acclimatization, line 141...Instead of starting the sentence with "That time"....use "During this time"

In experimental design, line 159... Instead of "Rats" write "rats"

Similar  errors should be screened thoroughly.

2. In  Drugs and Chemicals, please include the vendor's information.

Specific Comments

  1. Gentamicin used by the authors was pure antibiotic or in commercial form. Please mention this in the manuscript.
  2. A critical issue with this study is the ambiguity in the doses of the supplements. Was these fed with regular diet? If yes how the doses administered were recorded? Author say daily 20gm of water spinach or 5ml red grape juice or certain probiotic quantity were given to the test groups. How did the author controlled the intake? It would be clear if author mention these in a precise manner in the methodology.

Author Response

Reviewer 2

Comments and Suggestions for Authors

The manuscript focuses on the nutritional benefits of red grapes, water spinach and certain probiotics against gentamicin induced nephrotoxicity. The objectives of the study are significant with regard to the emerging interest in nutraceuticals or functional foods in therapeutic strategies or adjuvant therapies.

Authors’ responses:

Thank you very much for reviewing our manuscript and providing your expert opinions on our manuscript. We have taken into careful consideration all of your comments and tried our level best to improve our manuscript according to your suggestions.

1.The minor issue in the manuscript is some grammatical errors; for example- 

In Material and methods, line 117....Remove "the" before "searching"

In animal collection & acclimatization, line 141...Instead of starting the sentence with "That time"....use "During this time"

In experimental design, line 159... Instead of "Rats" write "rats"

Similar errors should be screened thoroughly.

Authors’ responses:

Thank you very much for your kind suggestions regarding the typos and grammar errors in the manuscript. We have carefully revised the whole manuscript and eliminated all types of unintentional typos and grammatical errors.

  1. In  Drugs and Chemicals, please include the vendor's information.

Authors’ responses:

Thank you very much for your suggestion. We have added the information vendor’s information as follows: 

Commercial Gentamicin injection (80mg) was collected from a model pharmacy (Dhaka, Bangladesh). Laboratory standard Thiobarbituric acid, Trichloroacetic acid, Hydrochloric acid, 10% formalin, 0.9% saline, Phosphate buffer, Detergent, Uricase, Dichlorophenol sulphonate, Ascorbate oxidase, Peroxidase, Amino-antipyrine, Urease Enzyme reagent, EDTA, Sodium salicylate, Sodium nitroprusside, Alkaline hypochlorite, Sodium hypochlorite, Sodium hydroxide, Urea standard, Picric acid, Creatinine Standard, Nitrite standard solution, Griess reagent, and Alcohol were purchased from Sigma Aldrich (St. Louis, MO, USA) and used during the research. (Lines: 148-155)

Specific Comments

  1. Gentamicin used by the authors was pure antibiotic or in commercial form. Please mention this in the manuscript.

Authors’ responses:

Thank you very much for your question. We have used pure form to induce nephrotoxicity in the study.  

2. A critical issue with this study is the ambiguity in the doses of the supplements. Was these fed with regular diet? If yes how the doses administered were recorded? Author say daily 20gm of water spinach or 5ml red grape juice or certain probiotic quantity were given to the test groups. How did the author controlled the intake? It would be clear if author mention these in a precise manner in the methodology.

Authors’ responses: Thank you very much for your critical observation and we appreciate your questions & subsequent suggestions in this regard. We have addressed and mentioned all the required information in methodology part as follows:

  • Normal control (NC) group: This group of rats received no treatment/gentamicin and was given the standard laboratory diet.
  • GEN group: Rats were provided with the standard laboratory diet.
  • GEN + WS group: In this group, the laboratory diet was mixed with boiled water spinach in a 4:1 ratio. So, each rat received about 20g boiled water spinach per day for seven days.
  • GEN + RG group: Apart from the basal diet, each rat received 5mL of red grape juice for the entire study period (7days). The juice was provided through oral gavage feeding tubes to ensure each rat's similar amount of intake.
  • GEN + P4 group: Throughout the entire study week, 4×109 CFU was provided through oral gavage feeding tubes to each rat each day.
  • GEN + P8 group: For the whole study period, 8×109 CFU was given through feeding tubes to each rat per day. (Lines: 161-173)
